# A Pilot Study: Favorable Effects of *Clostridium butyricum* on Intestinal Microbiota for Adjuvant Therapy of Lung Cancer

**DOI:** 10.3390/cancers14153599

**Published:** 2022-07-23

**Authors:** Jing Cong, Chuantao Zhang, Siyu Zhou, Jingjuan Zhu, Chengwei Liang

**Affiliations:** 1College of Marine Science and Biological Engineering, Qingdao University of Science &Technology, Qingdao 266042, China; zhousiyu9875@163.com; 2Department of Oncology, The Affiliated Hospital of Qingdao University, Qingdao University, Qingdao 266100, China; qddxzhangct@qdu.edu.cn (C.Z.); zhujj123456@126.com (J.Z.)

**Keywords:** *Clostridium butyricum*, probiotic, intestinal microbiota, adjuvant treatment, effects

## Abstract

**Simple Summary:**

A previous study has reported that the intestinal microbiota plays important roles in drug efficacy and toxicity in response to anticancer treatment. *Clostridium butyricum* could increase intestinal beneficial bacteria and has been clinically used in many diseases. Therefore, we tried to assess the roles of *Clostridium butyricum* by examining the composition, structure, diversity, marked differences, and interactional network of intestinal microbiota, as well as the progression-free survival, overall survival, and adverse events. The results showed that *Clostridium butyricum* supplement made some favorable changes in intestinal microbiota, such as the higher total richness of the genus *Clostridium*, *Bifidobacterium*, and *Lactobacillus*, and no distinguishing opportunistic pathogenetic markers, as well as the reduction in adverse events. This makes probiotics be promising adjunctive therapeutic avenues for lung cancer.

**Abstract:**

Probiotics as medications have previously been shown to change intestinal microbial characteristics, potentially influencing cancer therapy efficacy. Patients with non-squamous non-small cell lung cancer (NS-NSCLC) treated by bevacizumab plus platinum-based chemotherapy were randomized to obtain *Clostridium butyricum* supplement (CBS) or receive a placebo as adjuvant therapy. Clinical efficacy and safety were assessed using progression-free survival (PFS), overall survival (OS), and adverse events (AE). Intestinal microbiota was longitudinally explored between CBS and placebo groups over time. Patients who took CBS had significantly decreased bacterial richness and abundance, as well as increased the total richness of the genus *Clostridium*, *Bifidobacterium*, and *Lactobacillus* compared to the placebo group (*p* < 0.05). Beta diversity and the interactional network of intestinal microbiota were distinctly different between CBS and placebo group. However, there were no significant variations between them in terms of microbial taxonomical taxa and alpha diversity. The potential opportunistic pathogen *Shewanella* was still detectable after treatment in the placebo group, while no distinguishing microbial markers were found in the CBS group. In terms of clinical efficacy, the CBS group had a significantly reduced AE compare to the placebo group (*p* < 0.05), although no significantly longer PFS and OS. Therefore, favorable modifications in intestinal microbiota and significant improvements in drug safety make probiotics be promising adjunctive therapeutic avenues for lung cancer treatment.

## 1. Introduction

Lung cancer is the most commonly diagnosed cancer and the most lethal cancer in the world [1,2]. Non-small cell lung cancer (NSCLC), including adenocarcinoma, squamous cell carcinoma, and large-cell lung cancer, accounts for approximately 80–85% of lung cancer cases and is frequently diagnosed at an advanced stage. Platinum-based chemotherapy is considered the standard for the preferred frontline treatment to have improved survival and quality of life for NSCLC patients [3]. In recent years, the advent of targeted therapy and immune checkpoint inhibition has brought in great gains in patient outcomes [4,5]. Bevacizumab, a monoclonal antibody against vascular endothelial growth factor (VEGF), has been used to boost the chemotherapy efficacy in patients with non-squamous NSCLC (NS-NSCLC) [6,7]. Bevacizumab in combination with carboplatin and paclitaxel or other chemotherapy doublets had been shown to improve both progression-free survival (PFS) and the objective response rates of NS-NSCLC patients in phase 3 clinical studies [7,8]. Bevacizumab has a relatively favorable safety profile, with few adverse events (AEs) observed in a phase 4 study named SAiL [9]. The National Comprehensive Cancer Network (NCCN) guideline has recommended bevacizumab as a treatment for NS-NSCLC, either alone or in combination with chemotherapy [10,11]. Despite the demonstrated benefits of bevacizumab combined chemotherapy regimens for advanced NS-NSCLC, some patients still experience a few side effects, such as anorexia, fatigue, rash, diarrhea, neuropathy, bowel perforation, thrombosis, etc. [12,13].

It has been proven that intestinal microbiota has the ability to modulate cancer treatment [14,15,16]. It can directly affect the host’s metabolism and immune system or it can convert a drug to modify its pharmacodynamics [17]. High inter-individual variability in the composition of intestinal microbiota makes the different responses to a specific medicine in different individuals [18,19]. Probiotics as medications have the ability to alter intestinal microbial characteristics, including mucosal barrier function, immune responses, and gastrointestinal motility [20]. *Clostridium butyricum*, a butyrate-producing bacterium found in a variety of environments, has been widely and safely used as a probiotic for decades [21,22]. *C. butyricum* could increase intestinal beneficial bacteria, particularly *Lactobacilli* and *Bifidobacteria* [23,24]. Antimicrobial-associated diarrhea, inflammatory bowel disease, acute pancreatitis, autoimmune diabetes, and cancer have all been treated with *C. butyricum* without serious side effects [25,26,27,28,29]. According to the study by Tomita et al., *C. butyricum* was considered to play a favorable role in the therapeutic efficacy of immune checkpoint blockade in patients with lung cancer, which is closely associated with significantly longer PFS and overall survival (OS) [29].

In the context of this, we hypothesize that *Clostridium butyricum* supplement (CBS) will influence the treatment efficacy of patients with advanced NS- NSCLC treated with bevacizumab plus paclitaxel and carboplatin in our cohort study. Therefore, we try to assess the roles of *C. butyricum* by examining the composition, structure, diversity, marked differences, and interactional network of intestinal microbiota, as well as the clinical PFS, OS, and AE. Furthermore, our study tended to focus not only on the pre-treatment and post-treatment, but also on the dynamic variations of intestinal microbiota during the overall treatments. It will provide better understandings of how CBS influences the changes in intestinal microbiota following bevacizumab plus chemotherapy treatment. In addition, a specific probiotic supplement as a potential complementary treatment in combination with anticancer therapy will be worth exploring in future.

## 2. Materials and Methods

### 2.1. Patients and Medications

The cohort study enlisted the participation of twenty-one patients from the Affiliated Hospital of Qingdao University (Figure 1, Table 1 and Appendix A). All patients were histologically confirmed with locally advanced (stage IIIB) or metastatic (stage IV) NS-NSCLC and had not experienced any anticancer treatment. Other criteria for inclusion were radiological detectable disease, an Eastern Cooperative Oncology Group (ECOG) performance status of 0 or 1, and a sufficient function of a major organ. Those with inflammatory bowel disease, irritable bowel syndrome, and other intestinal illnesses that were treated with antibiotics, prebiotics, and other probiotics usage during the treatment were excluded. 

Patients were given bevacizumab at a dose of 15 mg/kg every three weeks plus paclitaxel and carboplatin until disease progression or intolerance. Among these NS-NSCLC patients, they were randomly divided into the CBS and placebo groups, named T and C. During the treatment, nine patients in T received *C. butyricum* (6 pills/day, 420 mg/pills, viable count ≥6.3 × 10^6^ CFU), while the remaining twelve patients in C received a placebo (Figure 1). The placebo group had similar packaging and received only starch and glucose—the medium in which the *C. butyricum* strains were removed. The probiotic preparation and placebo were supplied by Qingdao Donghai Pharmaceutical Co., Ltd. According to the manufacturer’s recommendations, the products were stored at 4 °C before and after being dispensed to parents. All of the participants in this study were local residents of Qingdao city. Before the trial initiation, the informed consents, including fecal samples collection and microbial analysis, were obtained from each patient. This cohort study was approved by the Affiliated Hospital of Qingdao University Institutional Review Board. Radiological evaluation was assessed every six weeks according to the Response Evaluation Criteria in Solid Tumors (RECIST 1.1). The efficacy and safety of probiotics were assessed by PFS, OS, and AE. PFS was explained as the time from the patients enrolled in this cohort study until the first evaluated tumor progression or death from any cause. OS was calculated from the time of treatment until the time of death or the last follow up. A survival curve was plotted with the Kaplan–Meier method and compared with a log-rank test. Safety was assessed by the frequency and type of AE in each group (Table 2 and Appendix A).

### 2.2. Fecal Samples Collection

Self-sampled feces samples were taken in the morning before the start of each treatment and stored at −80 °C until the day of analysis. A total of 110 fecal samples were collected from NS-NSCLC patients who had been treated with bevacizumab, paclitaxel, and carboplatin. Except for the disease progression or intolerance, all NS-NSCLC patients were observed for at least 18 weeks, according to the sample-supplying policy in the informed permission. These samples were named C1/T1, C2/T2, C3/T3, C4/T4, C5/T5, and C6/T6 in group C and T, respectively. Dynamic changes of intestinal bacterial characteristics were evaluated and analyzed in all available samples by 16S rRNA gene sequencing.

### 2.3. DNA Extraction and Gene Amplicon Sequencing

Genomic DNA from fecal samples of NS-NSCLC patients was extracted through the DNA Stool Kit from Tiangen [30], and then purified with 0.8% agarose gels. DNA quality was assessed using a NanoDrop 1000 Spectrophotometer (Thermo Fisher Scientific, Waltham, MA, USA) and measured absorbance at the ratio of 260/280 nm and 260/230 nm. DNA concentrations were quantified by Quant-iT PicoGreen dsDNA Assay Kit via a microplate reader (FLx800, BioTek, Winooski, VT, USA). Finally, the DNA samples were stored at −80 °C in preparation for further sequencing.

The sequencing of the 16S rRNA gene was performed as previously described [31]. The V3–V4 hypervariable regions of the bacterial 16S rRNA gene were targeted using the universal primer sets 357 forward (5′-ACTCCTACGGGRSGCAGCAG-3′) and 806 reverse (5′-GGACTACVVGGGTATCTAATC-3′). The gene amplification reactions were carried out in triplicate. The thermal cycling conditions were that: initial denaturation at 94 °C for 3 min, 94 °C for 25 s for 10 cycles, 53 °C for 25 s, 68 °C for 45 s, and lastly an extension at 68 °C for 10 min. The PCR products were visualized using 2% agarose gel electrophoresis and quantified on a microplate reader using the Quant-iT PicoGreen dsDNA Assay Kit (Thermo Fisher Scientific). As previously described [32], sample libraries for sequencing were prepared according to the MiSeq TruSeq Nano DNA LT Library Prep Kit Preparation Guide (Illumina). Prior to the sequencing, the library was checked for joints using the Agilent High Sensitivity DNA Kit and Agilent Bioanalyzer. The library was then re-quantified using Promega QuantiFluor’s Quant-iT PicoGreen dsDNA Assay Kit, and the concentration of the library was no less than 2 nM. In the following, it was denatured with 0.2 N fresh NaOH to form a single chain for sequencing. Finally, bacterial DNA amplicons were sequenced for 2 × 300 bp paired-end by a 600-cycles MiSeq Reagent Kit V3 (Illumina).

### 2.4. Sequencing Processing

The raw reads of the 16S rRNA gene were collected in FASTQ format by the MiSeq. These data were submitted to the sequence analysis pipeline (http://zhoulab5.rccc.ou.edu:8080; accessed on 10 July 2019) for further analysis based on the Galaxy platform [33]. The detailed processing referred to the previous process [34,35]. The spiked PhiX reads and primer sequences are removed by the Btrim program [36]. Forward and reverse reads were combined into a whole sequence by FLASH [37]. Any joined sequences that were less than 245 bp in length or had an ambiguous base were discarded. Thereafter, the clean reads were clustered into operational taxonomic units (OTUs) using UPARSE at 97% identity [38]. The remaining sequences were stripped of singletons and chimeras. 12,000 sequences were randomly selected (resampled) to normalize each sample to the same overall read abundance. The original detected OTUs were used in the rarefaction analysis (Appendix A). The OTU taxonomic classification was performed by the Ribosomal Database Project Classifier with 50% confidence estimates [39].

### 2.5. Network Analysis

To identify the clusters (modules) of closely linked intestinal taxa, the phylogenetic correlation networks were established by random matrix theory (RMT) methods based on the online MENA pipeline (http://ieg4.rccc.ou.edu/mena/; accessed on 9 November 2019). We conducted the network analyses with samples of the pre-treatment and post-treatment (C1, T1, C6, and T6). We kept those taxa that accounted for more than 60% of the relative abundance of intestinal microbiota. We judged the phylogenetic ecological network to be robust if the Spearman’s correlation coefficient was >0.50 and *p* < 0.01. To compare different ecological networks, the same cutoff of 0.74 was applied to construct ecological networks. We only focused on bacteria that are strongly and intimately interacting with one other; therefore, the cut-off has biological significance. Finally, we identified the modules of intestinal taxa that strongly interact only if each module has at least five nodes (OTUs). The modularity property was described by the fast greedy modularity optimization. 

In the following, we constructed the plot with the among-module connectivity (*Pi*) and within-module connectivity (*Zi*) to investigate the effect of *C. butyricum* on the topological roles of individual nodes. In this study, previous roles [40], which are peripheral nodes (*Zi* ≤ 2.5, *Pi* ≤ 0.62), connectors (*Zi* ≤ 2.5, *Pi* > 0.62), module hubs (*Zi* > 2.5, *Pi* ≤ 0.62), and network hubs (*Zi* > 2.5, *Pi* > 0.62) were used to classify the nodes in this study. The threshold value of *Zi* and *Pi* were explained by the density landscape of the nodes and by the basins of attraction for the different node density plots [41].

### 2.6. Statistical Analysis

The alpha diversity of the intestinal microbial community was evaluated using the Shannon index and Simpson index. The differences in intestinal microbial composition in different groups were examined using non-parametric multivariate statistical tests of dissimilarity and hierarchical clustering analysis. The dynamic process of intestinal microbiota during the treatment was illustrated using principal coordinate analysis (PCoA) and Bray–Curtis dissimilarity index. The dissimilarity tests referred to the multiple response permutation procedure (MRPP) algorithms, analysis of similarity (ANOSIM), and permutational multivariate analysis of variance (ADONIS). Significant *p*-values related with microbial clades were identified using linear discriminant analysis with effect size (LEfSe). The differences were analyzed by an LDA score cutoff of 2.0. The phylogenetic correlation networks were mapped based on the Cytoscape software (v2.8.3). All statistical analyses were performed by R software package (v3.4.1), except for two-tailed unpaired *t*-tests by IBM SPSS statistic 19.0 to determine the significant differences of different groups.

## 3. Results

### 3.1. Dynamic Taxonomical Composition Changes of Intestinal Microbiota

We assessed the taxonomical composition of intestinal microbiota in 110 samples of 21 subjects at six time points (Figure 1). A total of 4,467,962 quality-filtered 16S rRNA gene sequences were acquired from all available samples, with an average of 40,618 ± 7,568 reads per sample (Appendix A). A total of 940 OTUs (190 genera, 111 families) were generated at the 97% similarity level, with an average of 221 ± 59 OTUs per sample (Appendix A). In the following, the fluctuations in dynamic microbial composition were explored. Before treatment initiation, the phylum Firmicutes, Bacteroidetes, and Proteobacteria dominated the fecal microbiome of both C and T (Appendix A). At the phylum level, microbial composition in both C and T remained consistent as the treatment proceeded (Appendix A). There were almost no significant changes between inter-group and intro-group (Appendix A). At the genus level, these genera were almost not significantly different between C and T (*p* > 0.05), except for *Ruminococcus*, *Coprococcus*, *Roseburia*, *Streptococcus*, and *Akkermansia* in several rounds of treatment (*p* ≤ 0.05, Appendix A). In addition, there were group trends in the composition of Firmicutes, Bacteroides, and Proteobacteria (Appendix A).

### 3.2. Dynamic Diversity Changes of Intestinal Microbiota

During the entire treatment, T showed a significantly lower total taxonomic richness and abundance than C (*p* < 0.05, Figure 2a). As for dynamic analysis, the group taxonomic abundance in T was significantly reduced than that in C (*p* < 0.05, Figure 2b). The Simpson and Shannon diversity of the intestinal microbiota from NS-NSCLC patients in C fluctuated more notably than those in T during the overall treatment (Figure 2c). However, there were almost no significant differences between inter-group and intra-group in response to each round of treatment (*p* > 0.05, Appendix A). Dissimilarity tests revealed that the intestinal microbiota was almost not significantly different between pre-treatment and post-treatment, as well as between C and T based on the non-parametric MRPP algorithms, ANOSIM, and ADONIS in NS-NSCLC patients (*p* > 0.05, Appendix A). In addition, we also examined the changes in the genus *Clostridium*, *Bifidobacterium*, and *Lactobacillus* (Appendix A). The results showed that the total richness of the genus *Clostridium* was significantly higher in T than in C (*p* = 0.003, Appendix A,). The total abundance, the group richness and abundance had a greater advantage in T than in C, however, there was no significant weakness between them. The Shannon and Simpson index of the genus *Clostridium* increased in Week 9 and Week 12, indicating that this might have been influenced by CBS based on a potential accumulated process. However, it was distinctly decreased in Week 15. The total richness of *Bifidobacterium* and *Lactobacillus* was significantly higher in T than in C (*p* = 0.006, Appendix A). The total abundance, group richness, and group abundance of *Bifidobacterium* and *Lactobacillus* also had a distinct advantage in T than in C during the treatment, but not significantly.

The beta diversity of intestinal microbiota varied more obviously in C than in T during the overall treatment (Appendix A). In C, the Bray–Curtis distance of intestinal microbiota from NS-NSCLC patients in W6 and W9 was significantly lower than that in W0 (Wilcoxon rank-sum test, *p* < 0.05, Appendix A), while intestinal microbiota in T exhibited no significant changes throughout the treatment (*p* > 0.05, Appendix A). The beta diversity evaluated by Bray–Curtis distance in T was significantly higher than that in C (*p* < 0.05, Appendix A), except for C1 vs. T1, and C5 vs. T5 (*p* > 0.05, Appendix A). 

### 3.3. Changes of the Whole Community Structure and Potential Biomarkers of Intestinal Microbiota

The longitudinal gradient samples of intestinal microbiota were grouped into two clusters (C and T) based on hierarchical clustering analysis (Appendix A). T, respectively, formed new sub-clusters, such as the sub-cluster 1-1 (T1, T2, T3) and sub-cluster 1–2 (T4, T5, T6). C was clustered in a distinct gradient in response to the treatment (Appendix A). Intestinal microbiota in NS-NSCLC patients between C and T had clearly overlapped, although there were still some discrepancies, according to principal coordinates analysis (PCoA) based on the Bray–Curtis distance (Figure 3). PCoA demonstrated the dynamic process that intestinal microbiota greatly deviated from its initial state after treatment, and did not return to it over time, particularly for the larger distance between C1 and C6 than between T1 and T6 (Figure 3).

In addition, we examined the intestinal microbial clade differences at the taxonomical level using LEfSe analysis to identify intestinal microbial responses with CBS during the treatment. We made the comparison between C1 and T1 as well as between C6 and T6 (Figure 4). At the family level, greater proportions of Lachnospiraceae, Burkholderiaceae, and Shewanellaceae were found in C1, while Enterococcaceae and Leuconostocaceae were richer in T1 (Figure 4a). At the genus level, T1 had a greater prevalence of *Enterococcus* than C1, while C1 had a higher prevalence of *Burkholderia*, *Roseburia*, and *Shewanell* than T1. After five rounds of CBS, T had no identifiable microbial markers. However, *Shewanella* and Shweanellaceae were still the dominant genus and family in C, respectively (Figure 4b).

### 3.4. Dynamic Changes of Phylogenetic Interactional Network of Intestinal Microbiota

Phylogenetic molecular ecological networks were constructed to explore the effects of CBS on microbial assemblages that potentially interact with intestinal niches during the treatment. We selected the C1, C6, T1, and T6 to illustrate the changes of bacterial interactions. The representative networks from NS-NSCLC patients were constructed to modules with more than six biological duplicates and no fewer than five nodes in available samples (Figure 5a). There were one, one, three, and two module(s) in C1, C6, T1, and T6 networks, respectively (Figure 5a). The taxa in T tended to co-occur (positive correlations, grey lines) rather than co-exclude (negative correlations, blue lines) (Figure 5a). More than 50% of the potential interactions were negative correlations observed in C. The negative correlations rose by 2.60% from C1 to C6, but dropped by 4.60% from T1 to T6. 

Furthermore, *Pi* and *Zi* parameter space could be partitioned into different sections (Figure 5b). It has provided detailed information on connectors and module hubs in Appendix A. The OTUs from about 98.0%, 77.5%, 100.0%, and 97.1% of C1, C6, T1, and T6, respectively, were peripherals with the majority of their links inside their modules. In C1, there was only one OTU acting as module hub (those highly linked to abundant OTUs in their own modules). Compared with C6 (14 OTUs), fewer OTUs playing as connectors (those highly connected to several modules) were found in T6 (2 OTUs). No network hub OTUs (serving as both module hubs and connectors) were found in the two groups.

## 4. Discussion

Bevacizumab in combination with platinum-based chemotherapy has been recommended in advanced NS-NSCLC, with encouraging effectiveness and acceptable tolerability [42,43,44,45]. However, still some individuals experienced minor therapeutic benefits from the treatment. Increasing evidence suggested a close relationship between intestinal microbiota and lung cancer treatment [46]. Changes in intestinal microbial composition and function, termed dysbiosis, influence lung health by metabolism, inflammation, and immune response [47,48,49]. The pharmacological effects of drugs, such as activation [50], inactivation [51], toxification [52], and other compound modifications, are strongly influenced by intestinal microbiota. In turn, drug metabolites could have an impact on intestinal microbial composition and function. Therefore, in our cohort study, we examined the changes of intestinal microbiota and clinical responses in NS-NSCLC patients treated with CBS, which has been used to prevent intestinal microbial disturbances, strengthen intestinal barrier function, lower inflammatory factors, and enhance the immune response in intestinal tumorigenesis and colitis [27,53].

Firstly, we examined the changes in dynamic microbial composition between C and T across the course of NS-NSCLC patients’ therapy. Before treatment initiation, fecal microbiome in both C and T was dominated by Gram-positive Firmicutes, Gram-negative Bacteroidetes, and Proteobacteria, which was consistent with previous studies in healthy individuals [54], indicating that there was probably no severe intestinal microbial dysbiosis at the baseline in the cohort study. Intestinal microbial composition in T was not greatly changed with the administration of *C. butyricum* at the taxonomical level (Appendix A). In a previous experimental colitis model in mice, *C. butyricum* did not change the composition of intestinal microbiota [27,55]. There were almost no significant changes between the pre-treatment and post-treatment at the taxonomical level (*p* > 0.05, Appendix A). It is suggested that CBS had no discernible effect on the intestinal taxonomical composition. 

Secondly, we analyzed the changes in dynamic microbial diversity between C and T during the course of the treatment. The richness and abundance distribution of various types of microorganisms populating the gut is termed as intestinal microbial diversity [56]. When compared to C, T with CBS had significantly lesser total richness and abundance, as well as the group abundance (*p* < 0.05, Figure 2). PFS and OS were shorter in T (174 ± 30, 415 ± 50) than in C (187 ± 38, 421 ± 60), although not significantly (*p* > 0.05, Table 2). A Kaplan–Meier survival analysis also demonstrated the same PFS and OS result (*p* = 0.312, *p* = 0.233, Appendix A). It seemed to be consistent with a previous study that the close relationship between clinical responses and taxonomic richness (abundance) [57]. In response to the anti-PD-1 immunotherapy of hepatocellular carcinoma, non-responders had lower taxonomic richness and gene counts than responders [57]. In addition, we found that a relatively higher advantage in the richness and abundance of the genus *Clostridium* from the total and group level was found in T than in C at the baseline and during the treatment. Moreover, there was a significantly higher total richness of the genus *Clostridium* in T than in C (*p* < 0.05, Appendix A), indicating that CBS had a positive role in influencing the genus *Clostridium* species. Previous study has found that *C. butyricum* could increase certain beneficial bacterial taxa such as *Bifidobacterium* and *Lactobacillus* [58]. A similar conclusion was also verified in our result that the CBS group had significantly higher richness of *Bifidobacterium* and *Lactobacillus* than placebo group (*p* < 0.05, Appendix A). 

Although the Shannon and Simpson diversity of intestinal microbiota in T seemed to be more stabilized than in C (Figure 2c), there were almost no significant differences between T and C (*p* > 0.05, Appendix A). We speculated that it was probably attributed to the previous well-reported stability and resilience of individual characteristic of human intestinal microbiota [59,60]. We also examined the Shannon and Simpson diversity of the genus *Clostridium* (Appendix A). The Shannon and Simpson index of the genus *Clostridium* in T was distinctly increased in Week 9 and Week 12 but decreased in Week 15, indicating a potential accumulated effect of *C. butyricum* and chemotherapeutic drugs. Furthermore, throughout several rounds of treatments, a significantly higher value in beta diversity was found in T than in C (*p* < 0.05, Appendix A), suggesting that CBS exacerbated the disparity in intestinal microbiota between C and T. As a result, CBS could have different effects on the dynamic variation characteristics of intestinal microbial diversity during the platinum-based chemotherapy. 

Thirdly, we explored the differences between C and T in terms of community structure and microbial markers. Despite the fact that the overall samples had a lot of overlap, the core points of each group between C and T were clearly distinguished (Figure 3). In particular, all of the bacterial samples were constantly changing (Figure 3). We speculated that the differences between C and T were probably attributable to changes in immunological status and dietary behavior in response to treatment. Previous studies reported that adaptive immunity in response to cancer treatment could shape the colonic microbiome, which supports our speculation [61]. In addition, we compared group C and T to identify the intestinal microbial markers based on the LEfSe. In C1, the dominant genera were *Roseburia*, *Burkholderia*, and *Shewanella*, whereas it was *Enterococcus* in T1. *Enterococcus* spp. that was the most common cause of health-care-associated infections, and they were known for their adaptability to survive in harsh conditions [62]. For example, *Enterococcus faecalis* and *Enterococcus faecium* had intrinsic resistance to common antibiotics [63]. Elevated levels of *Enterococcus* spp. in intestinal microbiota were reported to be associated with lung cancer, making them potential biomarkers for lung cancer [64]. *Roseburia* spp. could produce short chain fatty acids, particularly butyrate, influencing intestinal motility, anti-inflammatory effects, and immunity maintenance [65]. Aside from *Burkholderia mallei*, numerous *Burkholderia* species were opportunistic pathogens capable of causing illness. Moreover, most *Burkholderia* spp. exhibited a modified lipopolysaccharide that resulted in intrinsic polymyxin resistance [66]. *Shewanella* spp. were mostly isolated from aquatic environments, however, they were recently linked to disease syndromes and multidrug resistance. According to a previous case, *Shewanella* spp. was associated with lung and bloodstream infections in cancer patients [67]. Generally, these bacteria are opportunistic pathogens that probably influence those immunocompromised people [68]. This is comparable to the fact that the majority of our recruited NS-NSCLC patients had a compromised immune system. It is suggested that intestinal microbiota of these recruited NS-NSCLC patients possessed possible opportunistic pathogens at the baseline, although not distinct at the phylum level (Appendix A). After five treatments, *Shewanella* remained the most prominent genus in C, while no marked taxa were found in T. It is noteworthy that the AE (frequency and type) in T (9 ± 1, 7 ± 1) was significantly lower than in C (29 ± 4, 16 ± 1) across the overall treatment (*p* < 0.05, Table 2 and Appendix A), especially in serious AE with T (11.1%) lower than C (16.7%). Therefore, we speculated that CBS had significantly improved intestinal microbiota with no distinguishing opportunistic pathogenetic markers and reduced the adverse events. 

Finally, we compared the interactional networks between C-enriched and T-enriched species. T-enriched species had a larger proportion of significant positive-correlation pairs than C-enriched species (Figure 5a). The positive-correlations had increased by 4.60% from T1 to T6. In comparison with T-enriched species networks, C probably had more fiercely competitive correlations among module species, indicating that their core microbiota was susceptible to change. In addition, higher connectors were found in C6 than in T6 (Figure 5b), suggesting that these increased nodes were probably inclined to connect with other modules. This further indicated that CBS in NS-NSCLC patients could provide strong support for the development of an intestinal-microbiome-modulation scheme during treatment [57].

This cohort study provides an understanding of changes in the intestinal microbiota from NS-NSCLC patients with CBS in response to anticancer treatment. Associated with the clinical symptoms, the improvement of patients with lung cancer is mainly based on the reduced AEs to improve the quality of life. In general, the analysis of fecal samples between the CBS group and the placebo group is not significant as we expected. We try to consider that the toxic side effects of the drugs potentially outweighing the effects of CBS. However, our cohort study comes with limitations. First, the changes may vary due to clinical context. We mainly focus on the microbial signature rather than the clinical outcomes. Second, since only 21 individuals were enrolled in our study, and were divided into two groups, further data from more participants are needed to reduce the inter-individual differences. A larger cohort will provide a better chance of avoiding the potential confounding factors. However, even with a small cohort, highly rigorous and validated results were drawn from two groups. Third, as only one dose of *C. butyricum* was used in the study, we cannot confirm the effects of CBS with different doses and the optimal dosing regimen. Fourth, aside from the structure, composition, and diversity of intestinal microbiota in influencing treatment responses, microbiota-derived metabolites, such as short chain fatty acids, should be good candidates for further investigation. In the future, the molecular mechanisms of *C. butyricum* regulating intestinal microbiota in NS-NSCLC patients treated with bevacizumab plus platinum-based chemotherapy will also be explored.

## 5. Conclusions

Rising lung cancer mortality represents the urgent need to develop innovative medicines and identify the optimal treatments for lung cancer patients. Here, we investigated the *C. butyricum* effects on dynamic changes of intestinal microbiota and clinical efficacy for a cohort of 21 NS-NSCLC patients in response to bevacizumab plus platinum-based chemotherapy. To some extent, we found that *C. butyricum* supplement made some favorable changes in significantly improving intestinal microbiota, such as the higher total richness of the genus *Clostridium**, Bifidobacterium* and *Lactobacillus*, and no distinguishing opportunistic pathogenetic markers, as well as reducing the adverse events. However, these results may suggest a limited change/improvement in intestinal microbiota and clinical efficacy. Aside from expanding the cohort, improving the depth of sequencing, considering the intestinal microbial metabolic factors, promising therapeutic avenues such as “design targeted probiotics” are probably used to effectively regulate intestinal microbiota in combination with anticancer treatment to reduce the individual heterogeneity and achieve the optimal clinical efficacy for lung cancer patients.

## Figures and Tables

**Figure 1 cancers-14-03599-f001:**
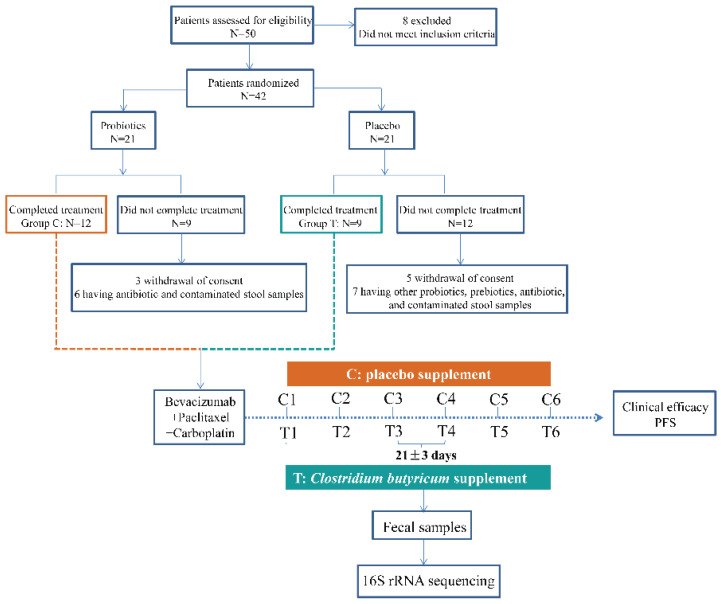
Study design. Twenty-one non-squamous non-small cell lung cancer (NS-NSCLC) patients were recruited for the cohort study. Fecal samples from these NS-NSCLC patients with placebo and *Clostridium butyricum* supplement (group C and T) were collected prior to each treatment.

**Figure 2 cancers-14-03599-f002:**
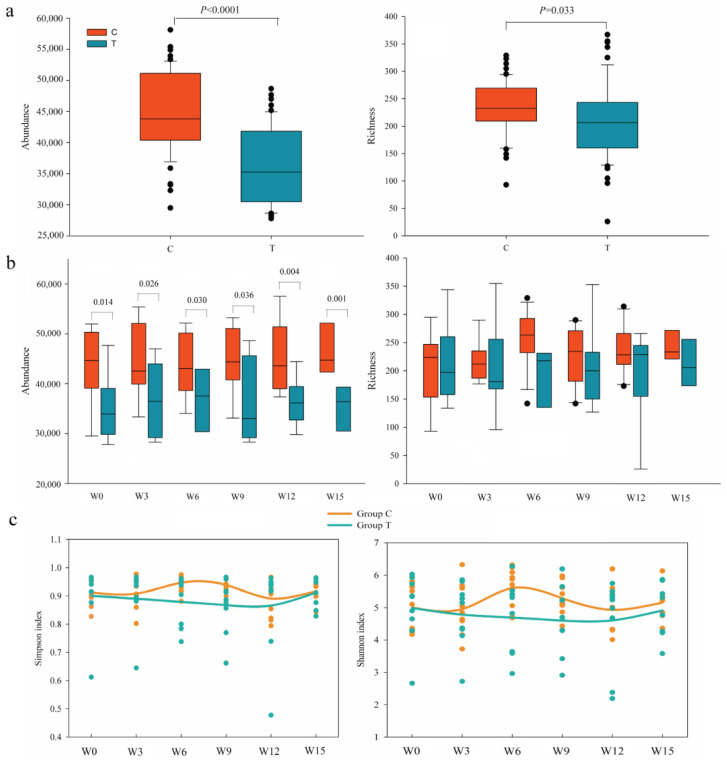
Differences in microbial composition between C and T. (**a**) The total abundance (left) and richness (right). (**b**) The group abundance (left) and richness (right). (**c**) Shannon and Simpson diversity.

**Figure 3 cancers-14-03599-f003:**
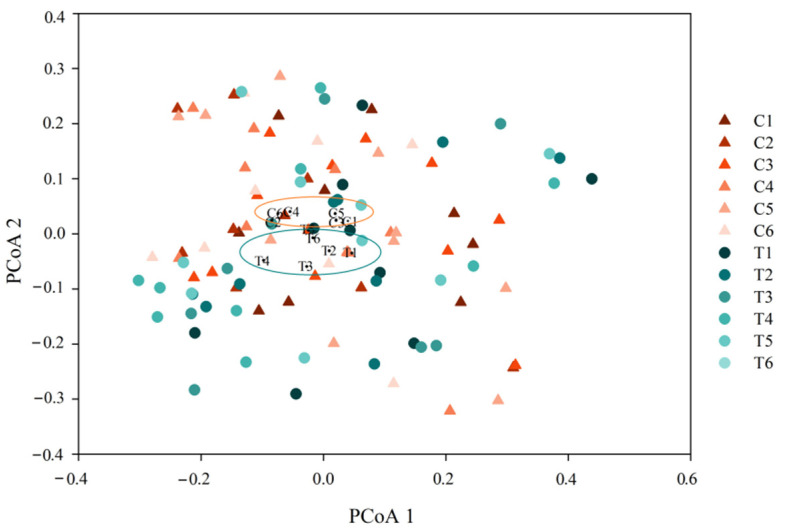
Principal coordinate analysis of intestinal microbiota in NS-NSCLC patients between C and T for each treatment.

**Figure 4 cancers-14-03599-f004:**
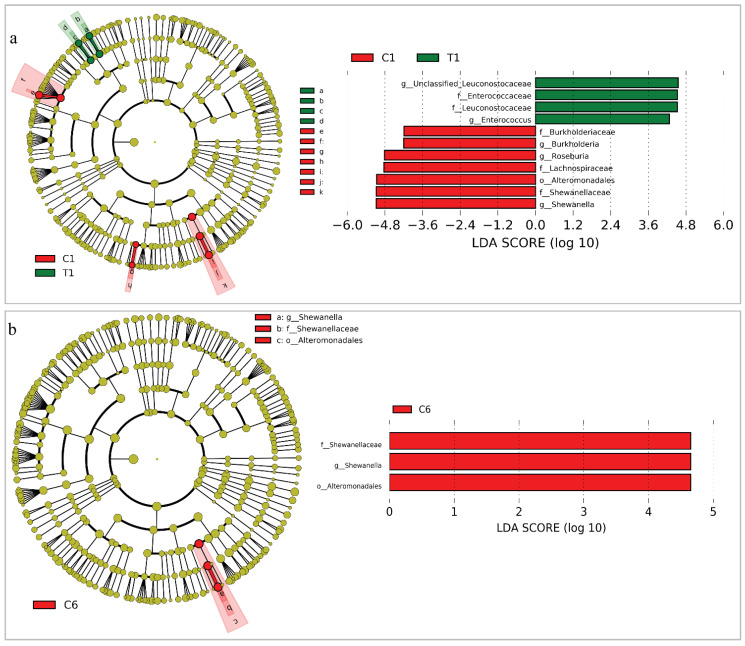
LEfSe analysis using Kruskal–Wallis test (*p* < 0.05) with LDA score > 2.0 and cladogram representation of the differentially abundant taxa between C1 and T1 (**a**), and between C6 and T6 (**b**). The size of each node means their relative abundance. The yellow taxa mean no significant difference between the two groups whilst, and the red and green taxa mean significant difference between the two groups.

**Figure 5 cancers-14-03599-f005:**
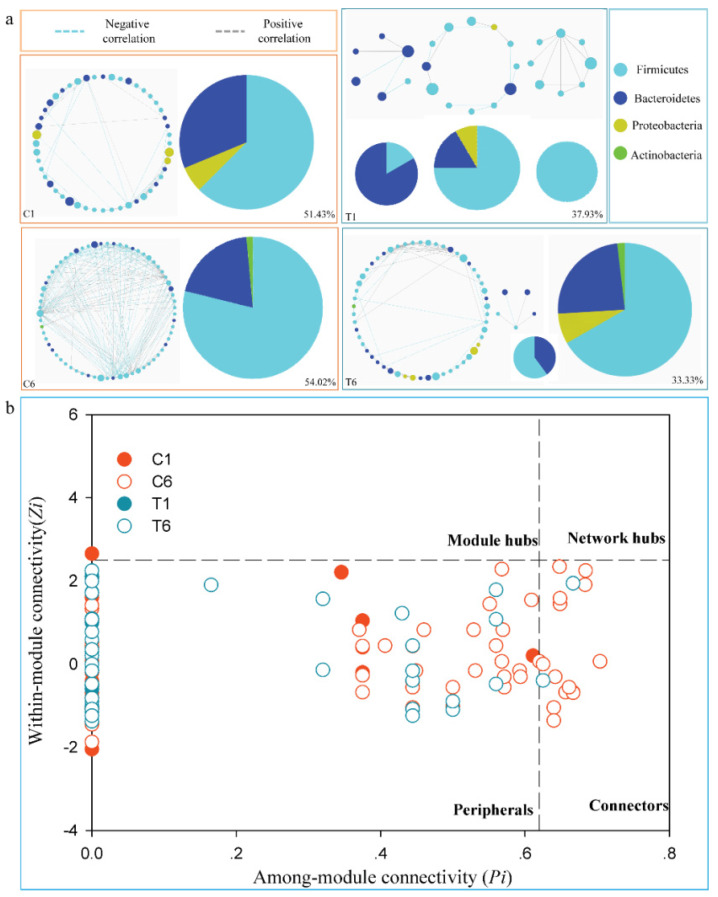
Dynamic changes of the phylogenetic interactional network. (**a**). Network interactions of intestinal microbiota in NS-NSCLC patients between C and T. Node colors mean different phyla; pie charts mean the module composition. The blue links indicate the negative correlations between two phyla, and the grey links indicate the positive correlations. The number means the ratio of negative links accounting for the total links. (**b**). *Z*–*P* plot showing the distribution of OTUs based on their topological characters.

**Table 1 cancers-14-03599-t001:** Clinical characteristic of 21 NS-NSCLC patients.

Clinical Parameter	Group T	Group C	*p* Value
Age			0.203
≥65	2	6	
<65	7	6	
Gender			0.204
Man	5	10	
Woman	4	2	
BMI			
≥24	3	4	0.596
<24	6	8	
Smoking			0.204
Yes	5	10	
No	4	2	
Stage			0.448
Ⅲ	1	3	
Ⅳ	8	9	

**Table 2 cancers-14-03599-t002:** Differences of clinical efficacy and safety between C and T.

Clinical Parameter		Group T	Group C	*p* Value
Efficacy	Progression-free survival (days)	174 ± 30	187 ± 38	0.799
	Overall survival (days)	415 ± 50	421 ± 60	0.941
Safety	Frequency of adverse events	9 ± 1	29 ± 4	0.001
	Type of adverse events	7 ± 1	16 ± 1	<0.001
	Serious adverse events	1/9 (11.1%)	2/12 (16.7%)	0.735

## Data Availability

The fecal microbial 16S rDNA sequences were deposited to the sequence read archive. The Bioproject accession for the 16S sequencing data is PRJNA574542 (https://dataview.ncbi.nlm.nih.gov/object/PRJNA574542, accessed on 26 May 2022).

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
