# Peer review of "A Pilot Study: Favorable Effects of Clostridium butyricum on Intestinal Microbiota for Adjuvant Therapy of Lung Cancer"

_cancers, 2022, doi:10.3390/cancers14153599_

Round 1
Reviewer 1 Report
The manuscript, "A pilot study: Favorable effects of Clostridium butyricum on intestinal microbiota for adjuvant therapy of lung cancer" deals with an important and interesting subject o the role of gut microbiota in lung cancer outcomes and the possible role of probiotics in improving the same, however, the study lacks scientific merit. The study comprises only 21 patients and the authors resort to comparison between individual patients eg. comparing C1 and T1. The claim that in Principal component analysis, the T1, T2, T3 and T4,T5,T6 form sub-clustures is an over-stretch. There are multiple attributes which can influence PFS and OS when treated with combination of probiotics and chemotherapy like metabolic and immunological factors which have not been taken into consideration. The authors only look at the gut microbial population which is again limited by the depth of sequencing.
Author Response
Reviewer 1
The manuscript, "A pilot study: Favorable effects of Clostridium butyricum on intestinal microbiota for adjuvant therapy of lung cancer" deals with an important and interesting subject o the role of gut microbiota in lung cancer outcomes and the possible role of probiotics in improving the same, however, the study lacks scientific merit. The study comprises only 21 patients and the authors resort to comparison between individual patients eg. comparing C1 and T1. The claim that in Principal component analysis, the T1, T2, T3 and T4, T5, T6 form sub-clustures is an over-stretch. There are multiple attributes which can influence PFS and OS when treated with combination of probiotics and chemotherapy like metabolic and immunological factors which have not been taken into consideration. The authors only look at the gut microbial population which is again limited by the depth of sequencing.
Response: Thank you for your letter and for the reviewer’ comments concerning our manuscript. Those comments are all valuable and very helpful for revising and improving our paper, as well as the important guiding significance to our researches. We have studied comments carefully and have made correction which we hope meet with approval. Revised portion are marked in red in the paper. The main corrections in the paper and the responds to the reviewer’s comments are as following.
First, we must clarify that the C1 and T1 are not the individual patients. We have listed the sampling timepoint of fecal samples from group C and T in Figure 1. The number behind of the group C and T represented the order of sampling. For example, C1 can be explained that these patients from the placebo group at the first sampling timepoint (W0), which was at the baseline. Intestinal microbiota from C1-1 can be explained as the collected samples from the first sampling of the first patient in placebo group. We are so sorry to that our inaccurate expression made you misunderstanding.
Second, we are also so troubled about the small sample size. In fact, we sampled a relatively big sample size in the beginning (see in Figure 1). However, many selected NSCLC patients were removed due to their personal reasons during the follow-up, including of taking antibiotics, other probiotics, and irregularly taking our probiotics. In addition, nonstandard sampling collection and contaminated samples by patients themselves, as well as insufficient fecal quantity made the sample number hard. So we had to abandon these samples, which were main reasons for small sample size. However, we try our best to reduce intra-group differences due to the small sample size. For example, we have sufficiently taken advantage of the same NSCLC patients before and after the therapy to explore the changes of intestinal microbiota. We collected 110 fecal samples in a longitudinal level for about 15 weeks. They were permanent residents who live more than five years in Qingdao. They were favored by very typical coastal dietary habit. Fecal samples were collected within three hours after defecation in the morning. During each time point of treatment, there are 9, 9, 12, 10, 10, 8 samples collected in the control group. And there are 9, 9, 8, 9, 9, 8 samples collected in treatment (prebiotic supplement) group. Therefore, we have no less than 8 samples as the experimental replications to make the significance analysis. The 8 sample replications in each treatment are enough to meet the ecological network analysis. In addition, we also found some studies with the small sample size, such as the article “Long-Term Oil Contamination Alters the Molecular Ecological Networks of Soil Microbial Functional Genes”, which was published in Frontiers in Microbiology. Twenty soil samples were collected from two groups. Another article is “Role of Fusobacteria in the serrated pathway of colorectal carcinogenesis”, which was published in Scientific reports. Eight tubular adenoma patients, ten sessile serrated adenoma patients and eight colorectal cancer patients were recruited in this study. Another article is “Gut microbiome affects the response to anti-PD-1 immunotherapy in patients with hepatocellular carcinoma”, which was published in Journal of Immunotherapy of Cancer. A cohort of eight patients was included in this study. However, it does not change the fact of the small sample size, especially for the human individual differences. Furthermore, although there were no totally significant changes in intestinal microbiota and clinical outcomes with prebiotics supplement, it is the fact in the real world, which has been examined by our data analysis.
Third, the sub-cluster 1-1 (T1, T2, T3) and sub-cluster 1-2 (T4, T5, T6) were constructed by hierarchical clustering analysis at the genus level. Principal component analysis based on the bray-curtis distance showed the distribution of intestinal microbial composition. We mainly focused on the central value of each fecal microbiota to explore the distance from the baseline to treatment end based on the PCoA plot. All these were analyzed by the rigorous statistical analysis.
Fourth, we very agree with your opinion. There are multiple attributes which can influence PFS and OS when treated with combination of probiotics and chemotherapy like metabolic and immunological factors. We did not take the intestinal microbial metabolites (such as SCFAs) and immunological factors into consideration. On the one hand, these fecal samples were collected in 2018, which was only set to explore these changes based on the 16S rRNA analysis due to the research fund and the price of metabolites analysis, and these samples have been lost so that we can’t make it up; On the other hand, the clinical efficacy of these patients were mainly evaluated by CT imaging analysis during the treatment, so these immunological factors and tumor markers were measured in a unfixed time. For example, one patient may have a SD result based on the imaging result, and the clinician generally did not ask the patients to detect these immunological factors and tumor markers on this condition. Especially, the detection time was not same as the fecal sampling time. Therefore, it is difficult to correlate with each other.
Fifth, actually, we only look at the intestinal microbiota changes based on the 16S rDNA sequence. It is our shortcoming, however, we have made sufficient analysis based these sampling data. In addition, we have added the part about the changes of the genus Clostridium. We found that the total richness of the genus Clostridium was significantly higher in T than in C. The total abundance, the group richness and abundance had a more advantage in T than in C, however, there was not significant between them. The Shannon and Simpson index of the genus Clostridium had an increasing in Week 9 and Week 12, and a distinctly decreasing in Week 15, indicating that a potential accumulated effect of C. butyricum and chemotherapeutic drugs.

Reviewer 2 Report
Thank you for the opportunity to review the manuscript entitled “A pilot study: Favourable effects of Clostridium butyricum on intestinal microbiota for adjuvant therapy of lung cancer”. This study looks at the effect of supplementation with Clostridium butyricum while undergoing adjuvant therapy on progressions free survival, overall survival, and adverse events as well as the impact of the gut microbiome. The results show changes to the gut microbiome and reduction in adverse events (type and frequency) but no impact of progression free survival or overall survival. The study of novel therapies for lung cancer are urgently needed as the 5 yr survival rate for NSCLC is 16% and has remained unchanged over the last 3 decades. Therefore, this research is highly relevant and necessary. There are a few limitations of study I would like to highlight and offer some suggestions for improvement.
Firstly, the author showed that this probiotic supplement significantly decreased overall abundance and richness, both of which contribute to alpha diversity, along with the Simpson and Shannon indices. A healthy gut microbiome has not been conclusively defined however there is a consensus that a healthy gut microbiome is highly diverse and abundant. As there was no significant difference in the other indices but a decrease in richness and overall abundance, the conclusion that the supplement improved the gut microbiome from baseline is not support by the results provided. I would suggest rewording this conclusion to account for this, in alignment with the results presented.
Secondly, I couldn’t find the actual compound that was used as the placebo. For transparency, the readers need to know what the placebo compound was so we can determine if it was appropriate for this study design. I would suggest clearly stating the placebo compound the methods section.
Lastly, did the CBS treatment increase the abundance of Clostridium butyricum in the gut? The results state that there were no significant taxonomic changes in the gut after CBS treatment when compared to placebo but it would be highly relevant to show the abundance levels of this specific bacterial species before and after treatment, assuming the goal was to increase abundance of CB as the mode of action for improvement. I suggest the authors add this data to the results section and where necessary in discussion.
Author Response
Reviewer 2
Thank you for the opportunity to review the manuscript entitled “A pilot study: Favourable effects of Clostridium butyricum on intestinal microbiota for adjuvant therapy of lung cancer”. This study looks at the effect of supplementation with Clostridium butyricum while undergoing adjuvant therapy on progressions free survival, overall survival, and adverse events as well as the impact of the gut microbiome. The results show changes to the gut microbiome and reduction in adverse events (type and frequency) but no impact of progression free survival or overall survival. The study of novel therapies for lung cancer are urgently needed as the 5 yr survival rate for NSCLC is 16% and has remained unchanged over the last 3 decades. Therefore, this research is highly relevant and necessary. There are a few limitations of study I would like to highlight and offer some suggestions for improvement.
Response: Thank you for your letter and for the reviewer’ comments concerning our manuscript. Those comments are all valuable and very helpful for revising and improving our paper, as well as the important guiding significance to our researches. We have studied comments carefully and have made correction which we hope meet with approval. Revised portion are marked in red in the paper. The main corrections in the paper and the responds to the reviewer’s comments are as following.
Firstly, the author showed that this probiotic supplement significantly decreased overall abundance and richness, both of which contribute to alpha diversity, along with the Simpson and Shannon indices. A healthy gut microbiome has not been conclusively defined however there is a consensus that a healthy gut microbiome is highly diverse and abundant. As there was no significant difference in the other indices but a decrease in richness and overall abundance, the conclusion that the supplement improved the gut microbiome from baseline is not support by the results provided. I would suggest rewording this conclusion to account for this, in alignment with the results presented.
Response: Thank you for your suggestion. We have revised the conclusion and aligned with the other part of manuscript. We consider that there was no distinguishing opportunistic after the treatment in CBS group compared to the pre-treatment. Thus, we suppose that intestinal microbiome from the baseline probably was improved to some extent in CBS group. Now, we have revised the part. We have added the changes of the genus Clostridium, Bifidobacterium, and Lactobacillus in the part of the results, we have revised the conclusion. “To some extent, we found that C. butyricum supplement made some favorable changes in significantly improving intestinal microbiota, such as the higher total richness of the genus Clostridium, Bifidobacterium and Lactobacillus, and no distinguishing opportunistic pathogenetic markers, as well as reducing the adverse events. However, these results may suggest a limited change/improvement in intestinal microbiota and clinical efficacy. Aside from expanding the cohort, improving the depth of sequencing, considering the intestinal microbial metabolic factors, promising therapeutic avenues such as “design targeted probiotics” are probably used to effectively regulate intestinal microbiota in combination with anticancer treatment to reduce the individual heterogeneity and achieve the optimal clinical efficacy for lung cancer patients.”
Secondly, I couldn’t find the actual compound that was used as the placebo. For transparency, the readers need to know what the placebo compound was so we can determine if it was appropriate for this study design. I would suggest clearly stating the placebo compound the methods section.
Response: Thank you for your suggestion. We have revised the part of the placebo in the methods. “The placebo group had similar packaging and received only starch and glucose—the medium in which the C. butyricum strains were removed. The probiotic preparation and placebo were supplied by Qingdao Donghai Pharmaceutical Co., Ltd. According to manufacturer’s recommendations, the products were stored at 4 °C before and after being dispensed to parents.”
Lastly, did the CBS treatment increase the abundance of Clostridium butyricum in the gut? The results state that there were no significant taxonomic changes in the gut after CBS treatment when compared to placebo but it would be highly relevant to show the abundance levels of this specific bacterial species before and after treatment, assuming the goal was to increase abundance of CB as the mode of action for improvement. I suggest the authors add this data to the results section and where necessary in discussion.
Response: Thank you for your suggestion. Based on the 16S rDNA sequencing depth, we can try best to explore changes in intestinal microbiota at the genus level. Therefore, we have added the part about the changes of the genus Clostridium Bifidobacterium, and Lactobacillus in Figure S4 and Figure S5. The result part is that “In addition, we also examined the changes of the genus Clostridium, Bifidobacterium and Lactobacillus (Figure S4, Figure S5). The results showed that the total richness of the genus Clostridium was significantly higher in T than in C (P = 0.003, Figure S4,). The total abundance, the group richness and abundance had a more advantage in T than in C, however, there was not significant between them. The Shannon and Simpson index of the genus Clostridium had an increasing in Week 9 and Week 12, indicating that it might been influenced by CBS based on a potential accumulated process. However, it was distinctly decreased in Week 15. The total richness of Bifidobacterium and Lactobacillus was significantly higher in T than in C (P = 0.006, Figure S5). Total abundance, group richness, and group abundance of Bifidobacterium and Lactobacillus also had a distinct advantage in T than in C during the treatment, but not significantly”. The discussion part is that “In addition, we found that a relatively higher advantage in the richness and abundance of the genus Clostridium from the total and group level was found in T than in C at the baseline and during the treatment. Especially, there was significantly higher total richness of the genus Clostridium in T than in C (P < 0.05, Figure S4), indicating that CBS had a positive role in influencing the genus Clostridium species. Previous study found that C. butyricum could increase certain beneficial bacterial taxa such as Bifidobacterium and Lactobacillus. Similar conclusion was verified in our result that significant higher richness in CBS group than in placebo group.” And “We also examined the Shannon and Simpson diversity of the genus Clostridium (Figure S4). The Shannon and Simpson index of the genus Clostridium in T were distinctly increased in Week 9 and Week 12, and decreased in Week 15, indicating that a potential accumulated effect of C. butyricum and chemotherapeutic drugs. Furthermore, throughout several rounds of treatments, significantly higher value in beta diversity were found in T than in C Furthermore, throughout several rounds of treatments, significantly higher value in beta diversity were found in T than in C (P < 0.05, Table S8), suggesting that CBS exacerbated the disparity in intestinal microbiota between C and T. As a result, CBS could have different effects on the dynamic variation characteristics of intestinal microbial diversity during the platinum-based chemotherapy.”

Reviewer 3 Report
Summary
Treatment for lung cancer through bevacizumad and chemotherapy elicits various adverse effects in patients. This pilot study by Cong et al. aims to investigate whether addition of the probiotics C. butyricum affects gut microbiota and adverse events when given in combination with treatment for NS-NSCLC.
The results showed that treatment altered the gut microbiota and reduced the frequency and severity of adverse events, though it failed to increase survival. There are a few major and minor points of concern for this study that the authors should address.
Major points
- What was the placebo used? To what extent are the beneficial effects due to the CBS? On this note, what methods have the authors taken to confirm that C. butyricum has colonised the gut – if that is to be expected?
- The authors touch upon a potential mechanism through which CBS exerts protective effects (positive correlation pairs in interactional networks). However, this should be elaborated, especially with relation to potential links with the reduced adverse events (both in the local gastrointestinal system, but also in the periphery – blood, liver, skin etc). Is this through bacterial ligands/metabolites that enable the gut microbiota communities to ‘cooperate’? Are metabolites (e.g. short chain fatty acids) in stool/blood altered that may explain the protective effects (the authors should perform analysis if there are sufficient samples)? Were there improved gut barrier functions with CBS?
- Were dietary/lifestyle factors considered and how would they impact the interpretation of the current data? For example, the women in the study didn’t smoke, and smoking has an effect on the gut microbiota composition; does CBS have a different/better effect in non-smokers vs smokers?
Minor points
- [ln 55] Please elaborate on “a few side effects” for ease of reading
- [ln 59-61] Please revise and rephrase these 2 sentences. It is unclear what the authors are trying to express
- Table 2 (also in Table S2). “advent events” should be “adverse events”
- [ln 224] “intro-group” should be “intra-group”
- [ln 299] Delete “authors”
- [ln 371] “tara” should be “taxa”
Author Response
Reviewer 3
Summary
Treatment for lung cancer through bevacizumad and chemotherapy elicits various adverse effects in patients. This pilot study by Cong et al. aims to investigate whether addition of the probiotics C. butyricum affects gut microbiota and adverse events when given in combination with treatment for NS-NSCLC.
The results showed that treatment altered the gut microbiota and reduced the frequency and severity of adverse events, though it failed to increase survival. There are a few major and minor points of concern for this study that the authors should address.
Response: Thank you for your letter and for the reviewer’ comments concerning our manuscript. Those comments are all valuable and very helpful for revising and improving our paper, as well as the important guiding significance to our researches. We have studied comments carefully and have made correction which we hope meet with approval. Revised portion are marked in red in the paper. The main corrections in the paper and the responds to the reviewer’s comments are as following.
Major points
- What was the placebo used? To what extent are the beneficial effects due to the CBS? On this note, what methods have the authors taken to confirm that C. butyricum has colonized the gut – if that is to be expected?
Response:
(1) The placebo group had similar packaging and received only starch and glucose—the medium in which the C. butyricum strains were removed. The probiotic preparation and placebo were supplied by Qingdao Donghai Pharmaceutical Co., Ltd. According to manufacturer’s recommendations, the products were stored at 4 °C before and after being dispensed to parents. It has been added in the part of material and methods.
(2) We consider the beneficial effects of the CBS on the improved clinical efficacy (PFS and OS), the lower adverse events, as well as the favorable changes in intestinal microbiota (no distinguishing pathogenic microbial markers were found in CBS group, higher beneficial bacterial taxa Clostridium, Bifidobacterium, and Lactobacillus), which have been described in our revised manuscript.
(3) Actually, we did not take the detailed studies to confirm the colonization of C. butyricum in our pilot study. First, total 110 fecal samples were collected along the longitudinal time gradient in 2018. Considering our previous funding condition, so larger number of fecal intestinal microbiota was mainly explored by 16S rDNA amplicons sequencing. Compared to the shotgun metagenomic sequencing, insufficient sensitivity of 16S rDNA sequencing is not enough to detect the C. butyricum colonzation, which is also explained by previous study that using a combined culture-dependent and –independent approach utilizing 16S rDNA and shotgun metagenomic sequencing and qPCR validation readily identified the probiotics strains with high specificity (Zmora, N. et al, 2018). Therefore, we have to comprehensively analyze the 16S rRNA sequencing data to reflect the overall intestinal microbial changes. Second, inter-individual variations lead to the differences in probiotics colonization of the human GI Tract. Previous findings have pointed out that human consumption of probiotic strains results in universals shedding in stool but with highly individualized mucosa colonization patterns (Zmora, N. et al, 2018). However, it is not possible to take intestinal tissue to detect the colonization of C. butyricum from lung patients in our pilot study. Considering it, it is probably not accurate if the colonization of C. butyricum is detected from our fecal samples. Thus, we mainly focused on the clinical efficacy and changes of intestinal microbial community by CBS in lung patients treated with bevacizumab plus paclitaxel and carboplatin.
Reference: Zmora, N., Zilberman-Schapira, G., Suez, J., Mor, U., Dori-Bachash, M., Bashiardes, S., Kotler, E., Zur, M., Regev-Lehavi, D., Brik, R. B., Federici, S., Cohen, Y., Linevsky, R., Rothschild, D., Moor, A. E., Ben-Moshe, S., Harmelin, A., Itzkovitz, S., Maharshak, N., Shibolet, O., Elinav, E. (2018). Personalized Gut Mucosal Colonization Resistance to Empiric Probiotics Is Associated with Unique Host and Microbiome Features. Cell, 174(6), 1388-1405.e21. https://doi.org/10.1016/j.cell.2018.08.041.
- The authors touch upon a potential mechanism through which CBS exerts protective effects (positive correlation pairs in interactional networks). However, this should be elaborated, especially with relation to potential links with the reduced adverse events (both in the local gastrointestinal system, but also in the periphery – blood, liver, skin etc). Is this through bacterial ligands/metabolites that enable the gut microbiota communities to ‘cooperate’? Are metabolites (e.g. short chain fatty acids) in stool/blood altered that may explain the protective effects (the authors should perform analysis if there are sufficient samples)? Were there improved gut barrier functions with CBS?
Response: Thank you for your suggestions. The interactional network was constructed by intestinal phylogenetic microbial sequencing data based on the integral dataset from these participants in CBS or placebo groups. So network topological parameters have only one value in a sampling timeline based on all patients from each group. We also try to explore the correlations of the clinical efficacy (PFS, OS) and adverse events, and intestinal microbial data (abundance, diversity, network data). However, the clinical efficacy (PFS, OS) and adverse events are considered as an accumulated data, and intestinal microbial data from each sampling timeline is an instant data. Therefore, we did not take these analyses for further analysis. In the previous sampling time and study setting, we focused firstly on the clinical efficacy and adverse events by CBS in lung patients treated with bevacizumab plus paclitaxel and carboplatin, secondly on the changes of intestinal microbial community structure and composition. However, we did not consider the changes of intestinal microbial metabolites (e.g. short chain fatty acids) at that time, and unfortunately we have no enough and active samples to further explore the microbial metabolites. In the future study, we will consider that not only the structure and composition but also the function and metabolites. In addition, we added the changes of the genus Clostridium, Bifidobacterium, and Lactobacillus (Figure S4, Figure S5). The results showed that the total richness of the genus Clostridium, Bifidobacterium, and Lactobacillus were significantly higher in T than in C (Figure S4, Figure S5), which is in the revised manuscript.
- Were dietary/lifestyle factors considered and how would they impact the interpretation of the current data? For example, the women in the study didn’t smoke, and smoking has an effect on the gut microbiota composition; does CBS have a different/better effect in non-smokers vs smokers?
Response: Indeed, dietary and lifestyle factors could influence the intestinal microbiota. Here, we specially selected these lung patients from local residents of Qingdao city to reduce the regional disparity. In addition, certain lifestyles, such as smoking, actually have effects on the intestinal microbiota. In our study, we did not pay more attentions to these factors, because we consider that these lifestyles in individuals themselves remain relatively unchanged. We collected these fecal samples from the same individual on a longitudinal scale. We compared different timelines of intestinal microbiota from these same patients. To some extent, only these treatment drugs and CP were considered as new additive influencing factors. Therefore, we did not explore these detailed dietary and lifestyle factors separately in our pilot study. However, these factors cannot be ignored as an individual. In the future study, we will pay more attention to these influencing factors of overall and individual microbial changes.
Minor points
1.[ln 55] Please elaborate on “a few side effects” for ease of reading
Response: Thank you for your suggestion. We have revised it. “Despite the demonstrated benefits of bevacizumab combined chemotherapy regimens for advanced NS-NSCLC, some patients still experienced a few side effects, such as anorexia, fatigue, rash, diarrhea, neuropathy, bowel perforation, thrombosis, etc 12, 13.”
2.[ln 59-61] Please revise and rephrase these 2 sentences. It is unclear what the authors are trying to express
Response: Thank you for your suggestion. We have revised it. “High inter-individual variability in the composition of intestinal microbiota makes the different responses to a specific medicine in different individuals”.
3.Table 2 (also in Table S2). “advent events” should be “adverse events”
Response: Thank you for your suggestion. We have revised it.
4.[ln 224] “intro-group” should be “intra-group”
Response: Thank you for your suggestion. We have revised it.
5.[ln 299] Delete “authors”
Response: Thank you for your suggestion. We have revised it.
6.[ln 371] “tara” should be “taxa”
Response: Thank you for your suggestion. We have revised it.

Round 2
Reviewer 1 Report
The authors mention that the original samples are lost and hence no experiment/analysis can be done further. In that case, there is no point of evaluating a revised manuscript. Also, argument of previous papers published with small sample sizes cannot be justified, the field is rapidly growing.
Reviewer 3 Report
Thank you for the revisions. The discussion is much more integrated and I understand the lack of samples for conducting further experiments.